# A Uniform Magnetic Field Generator Combined with a Thin-Film Magneto-Impedance Sensor Capable of Human Body Scans

**DOI:** 10.3390/s22093120

**Published:** 2022-04-19

**Authors:** Tomoo Nakai

**Affiliations:** Industrial Technology Institute, Miyagi Prefectural Government, Sendai 981-3206, Japan; nakai-to693@pref.miyagi.lg.jp

**Keywords:** magnetic thin film, magneto-impedance sensor, nondestructive inspection, human body

## Abstract

A detection system for magnetic inclusions of large bulk, such as that of a whole human body, is proposed in this paper. The system consists of both a uniform magnetic field generating apparatus capable of the insertion of a whole human body and also of a high-sensitivity magnetic sensor array installed in the strong magnetic field. The system can detect the magnetic inclusion simultaneously through its magnetization, which is advantageous for detecting low-remanence magnetic materials, such as a cluster of nanoparticles. The thin-film magneto-impedance sensor was reported to be capable of tolerating strong magnetic fields of more than 3000 Gauss (0.3 T) in the substrate’s normal direction and can retain its sensitivity even in strong fields. Through a combination of both uniformity of strength and the placement of its directionally aligned, static magnetic field in a particular measurement area and its array of single-dimensional thin-film magneto-impedance sensors, it was reported that it can estimate a magnetic sample’s 3D position by using a simple equation. The aim of the system developed in this study is to nondestructively detect a cluster of magnetic nanoparticles in a human body and also to detect the position and the concentration of the clustered magnetic particles. In this paper, a prototype system consisting of a magnetic field generator with an area of W500 mm × L400 mm × H210 mm and a uniform magnetic field of 370 Gauss (37 mT) is reported. It also reported that the thin-film magneto-impedance sensor installed in the system verified the detection of 2 mm × 1 mm small ellipsoidal magnetic chips at a distance of 27 mm from the sensor element.

## 1. Introduction

A thin-film magneto-impedance sensor [1,2,3,4,5,6] is useful for detecting magnetic materials in industrial and medical products [7,8,9,10,11,12,13,14,15]. The magneto-impedance sensor has the ability to detect a cluster of magnetic nanoparticles from a particular distance [13,16,17,18]. A system which can detect soft, small magnetic pieces using a thin-film magneto-impedance sensor within a strong static field is proposed, and it can detect submillimeter-sized magnetic particles nondestructively [19,20,21,22,23]. Such particles are detected by the sensor simultaneously with magnetization using a particular magnetic structure installed with sensors. The thin-film sensor is installed in a strong magnetic field and retained its sensitivity [21]. The reason the sensor is able to detect a small particle in the strong magnetic field is the tolerance against the surface normal field of the thin-film sensor, which is due to the demagnetizing field along the thickness direction of the sensor. A system which can estimate the position of a magnetic contaminant is proposed using a combination of a uniform vertical magnetic field and an array of in-plane, single-directional magnetic field sensors [23]. It can estimate both the 3D position and also the value of the magnetization. In the proposed system, the sensing direction of the sensor, which is the in-plane direction of the thin-film sensor, is set in the perpendicular direction to the applied strong magnetic field. Position and size estimation are carried out using the measured signal waveforms obtained by the nearer two sensors while feeding the sample linearly to the measurement area.

In this study, a uniform and strong magnetic field generator for a particular large volume applicable to a human body scan is proposed, in which both the position and the volume of the magnetic concentration are possible to estimate. Recent progress in medical technology utilizes magnetic nanoparticles that can be introduced into the human body. Resovist^®^, a kind of super paramagnetic iron oxide (SPIO) nanoparticle, is well known for its use in contrast-enhancement for magnetic resonance imaging (MRI). It consists of a mixture of Fe_2_O_3_ and Fe_3_O_4_ nanoparticles having paramagnetic properties. It has a bio-compatibility with the human body, thus many trials of medical applications using iron oxide magnetic nanoparticles, such as those for magnetic separation [24], magnetic drug delivery [25], magnetic targeting [26], and hyperthermia for cancer treatment [27], have been undertaken recently. Detection of locally concentrated magnetic nanoparticles, such as those present in a cancer or a lymph node, is becoming important for the purpose of preoperative detection of cancer metastasis. A high-sensitivity magnetic field sensor combined with such a field generator is a candidate for use in such a medical diagnostic test.

## 2. Concept of the Measurement System

In this section, a magnetic field generator which is capable of applying a whole human body scan with a combination of a thin-film magneto-impedance sensor for detecting the magnetic leakage field from a cluster of magnetic nanoparticles is proposed, and the concept of this system is introduced.

Figure 1 shows a schematic image of the proposed inspection system. It consists of a soft, rectangular magnetic core and magnets with opposite magnetic poles facing each other from each inner side of the core. The magnetic core works as a closed-loop magnetic field in order to make the inner magnetic field greater. On the inner surface of the magnets there are magnetic flux homogenization plates that consist of soft magnetic material of high-saturation magnetic flux density. Due to the effect of this homogenization plate, it is possible to compose the magnet on one side of several separated magnets that make a uniform magnetic field. After this, the proposed magnetic structure is able to make a uniform field in strength and of a constant directional field in the space between the upper and lower magnets. This measurement system was developed for the application of a small-sized work piece [22], and we have also proposed a method for estimating the position of a magnetic particle [23].

Figure 2 is an image of the measurement of a human body using the proposed inspection system. A cluster of magnetic particles is magnetized by the generated uniform and constant directional field, and an array of magnetic sensors which are set inside the magnetic field is able to detect the field coming from the magnetized particle cluster. This system offers the advantage of detecting low-remanence magnetic particles due to the fact that the sensor measures the magnetic particles simultaneously with the magnetization inside the strong magnetic field. In this system, the magnetic sensor is a thin-film sensor in which the sensing direction is along the direction of the film plane, and which sets it in the uniform and constant directional strong magnetic field with the sensing direction set perpendicular to the magnetic field. The sensor works by retaining high sensitivity even in such a strong magnetic field due to the property of tolerance against the surface of the normal magnetic field. Based on our previous study, the position estimation of the small magnetic piece was realized using a measurement system which possesses the same structure shown in Figure 2.

The study induced an analytical equation for estimating 3D position based on the equation for a magnetic field generated by a magnetic dipole. The resultant estimation shows that the height of the small piece, *h*, is calculated by the equation *h* = 2 × XSP when the chips are run through just above the sensor, where XSP is a position of extreme value of the measured magnetic waveform obtained by the sensor. Figure 3 explains the relationship between the feeding height of the magnetic dipole and its measured waveform of magnetic flux density obtained by the sensor, which is shown in Ref. [23]. The waveform indicates variation as a function of the feeding position of the dipole at a particular height. It shows that the relationship between the feeding height *h* and the position of extreme point XSP is *h* = 2 × XSP. The position of extreme point XSP is an important parameter for position estimation [23]. Then, the longitudinal dimensions of the measurement area must have a larger value, which is the value of *h*’/2, where *h*’ is the height of the expected maximum measurement area of a magnetic piece. In this study, the height of the measurement area was assumed to be 225 mm, which is available from the human body. The average value of the chest depth of a Japanese male is reported to be 212 mm [28]. Owing to the height of the measurement area, the longitudinal dimension is assumed to be 112.5 mm as a minimum value, which is based on the demands of the height estimation. The dimensions of the prototype system were designed as 200 mm on each side, taking into consideration the effect of the magnetic field expanding in the fringe area. Because this dimension is needed for both the + and − directions, the actual longitudinal dimensions of the system were set at 400 mm and were in the feeding direction. In this study, the measurement area, which has a uniform vertical magnetic field, was designed as W700 mm × L400 mm × H225 mm. This longitudinal dimensions can be decreased when the sensor arrays are set on both sides of the measurement area, especially when they are set on the surface of the lower and upper sides of the magnetic homogenization plate.

In our proposed system, the magnetic sensor is a thin-film magneto-impedance sensor driven by 400 MHz high-frequency electric circuit [20]. The merit of this circuit was introduced by our previous article. It has the ability to realize a high sensitivity for driving the thin-film MI sensor, and also makes it possible to align the sensors in a suitable bias field position in the strong magnetic field of the measurement area. The MI sensor needs a particular value of the magnetic bias field, so it must be set by controlling the position and azimuthal angle of the sensor element within the strong field of the system. In our prototype system, a particular slight distribution of the vertical field, which has a partial vector along the sensor’s sensing direction, is generated and makes it possible to control the sensor bias point. The driver circuit was suitably designed for this purpose. An electrically adjustable attenuator is set in our circuit and can switch the operation mode from differential amplification to single-ended amplification by setting the attenuation value to the maximum. In this case, the signal output of this circuit indicates the value of the sensor impedance. This circuit is effectively used for adjusting the sensor position and direction by monitoring the sensor impedance value for the purpose of adjusting the bias point of the sensor operation.

Figure 4 shows a schematic explanation of the bias field adjustment. The sensor’s driver circuit, in which the attenuator is set as the maximum attenuation, outputs a signal that is negatively proportional to the sensor impedance. In this case, if the profile of the magnetic field distribution is known in advance, then it is possible to set the sensor in a suitable position. It is desirable to design the partial vector of the magnetic field to have a slight variation along the sensor’s sensing direction. In this study, the direction is matched to the feeding direction of the system. In this case, it is possible to adjust the sensor bias point by controlling the sensor position mechanically in the feeding direction. It is well known that high-frequency electric circuits which drive more than 100 MHz have difficulty setting an electrical switch in the circuit due to the demand of impedance matching. The proposed driver circuit has the advantage of this function in addition to realizing a high-sensitivity measurement using the MI sensor. The combination properties of an MI sensor and a magnetic field generator are shown in the latter part of this article.

## 3. Experimental Results

In this section, an actual fabrication of the proposed system is shown. Verifications of the structure of the generated magnetic field and also verification of the sensor installation set in the strong magnetic field with the sensing direction perpendicular to the strong magnetic field are shown here. 

### 3.1. Experimental Apparatus

The structure of the magnetic field generating apparatus which was designed and fabricated in this study is explained.

Figure 5 indicates the design layout of our proposed field generation apparatus. This is a magnetic field generator which is made of square sections of NdFeB magnets and magnetic homogenization plates. The left figure is of the side view and the right one is the plan view of the apparatus. It consists of a pair of units having the upper and lower ones with the opposite magnetic poles facing each other. One of the units has 18 magnets, with the dimensions of each magnet being W100 mm × L100 mm × H20 mm, to form a magnet array of 3 × 6 matrix layout. The magnet is a product that is being marketed. The magnetic field homogenization plate has dimensions of W740 mm × L450 mm × H12 mm. It was made of S45C soft magnetic steel. The distance between magnets was set as 20 mm width X and 45 mm length Y. The middle law of the magnets in the three laws were directly fixed on the homogenization plate, which is shown in the left side of Figure 5. The magnets laws on both sides had a particular distance from the plate for the purpose of making the in-plane directional bias field around the upper side of the directly fixed middle-law magnets. The position of the sensor array was actually a particular distance from the central position in order to control the sensor bias point at a suitable magnetic field value by controlling the sensor longitudinal position. The magnets of each law were fixed on a base plate made of S45C steel. This base plate has dimensions of W900 mm × L120 mm × H12 mm and was fixed on magnets on another side of the homogenization plate. The base plates were designed to be connected with the side steel walls of the apparatus to form a closed magnetic structure, which is the same structure of the magnetic core in Figure 1. In this study, the distance between the two homogenization plates was designed as 225 mm, and a uniform magnetic field with a slight distribution is generated between them.

Figure 6 shows a perspective photograph of the fabricated measurement system. This system has a moving table for carrying the measured work piece, such as a human body or industrial products. The materials used in this system were selected as non-magnetic materials, such as SUS304, aluminum and thermosetting resins. An electrical actuator and sliding rail units for feeding the carrying table were set just above the floor to minimize the magnetic effect on the measurement sensors. In this photo, there is a gaussmeter on the carrying table for verifying the generated magnetic field. A digital oscilloscope for measuring and memorizing the distribution of the magnetic field was set beside the apparatus. The measurement of the magnetic field profile was carried out using the automatic feeding table by positioning the hall probe in various width and height positions.

### 3.2. Magnetic Field Distribution

An experimental confirmation of the fabricated magnetic field distribution was carried out and reported here. It confirms the wide area uniformity of the magnetic field and also that the magnetic field is at the sensor position which is suitable for sensor operation.

Figure 7 shows a schematic of the measurement procedure of the magnetic field profile. In this measurement, magnetic flux density in Z-direction B_z_ was measured by the feeding Hall probe at a constant speed of 100 mm/s. The original point of the measurement coordinates was set in the middle of the surface of the lower homogenization plate. The measurement area ranges from −300 mm to +300 mm in the feeding Y-direction, from −500 mm to +500 mm in the width X-direction and from +38 mm to +174 mm in the vertical Z-direction. The position on the carrying table in which the Hall probe was set just above the surface was with *z* = 38 mm.

From here, the measurement results of the generated magnetic field by our proposed system are shown.

Figure 8 shows the measured magnetic field when the vertical position was *z* = 38 mm, which is just above the upper surface of the carrying table. The B_z_ suddenly rose up almost above the edge of the homogenization plate, which was *y* = ±225 mm, and there was an almost flat B_z_ area above the plate. The value at the center of the plate was 370 G (37 mT). The waving variation existed around the top of the table in which the variation range was lower than 2%. In this case, the B_z_ was apparently constant as a function of transverse position *x*.

Figure 9 shows the case of *z* = 93 mm, which is 20 mm lower from the vertical middle height. In this case, the value of the center position was the same value, B_z_ = 370 G (37 mT), as the previous result. The field value gradually decreased as it got closer to the edge of the homogenization plate. At the edge of the transverse range, *x* = ±300 mm, the value was 330 G (33 mT), which was 11% decreased from the center value. This was expected by the preliminarily carried out magnetic field simulation and was caused by the effect of magnetic flux line expansion outward from the strong area.

Figure 10 is the overlapping expression of the field variation as a parameter of vertical height *z*. The horizontal axis represents the feeding position Y, and the vertical axis represents the flux density B_z_. The transverse position was fixed at *x* = 0, which was on the center line. The parameter *z* ranged from *z* = 38 mm to *z* = 174 mm, which was within the distance between the homogenization plates, 225 mm. The result show that the vertical flux density was almost the same value independent of the height position. Of special note, the value of the central position was a constant 370 G (37 mT). A slight difference existed at around the fringe area of the homogenization plate. This result shows a verification that our proposed structure is able to generate an almost constant vertical magnetic field of 370 G (37 mT) within a W500 mm × L400 mm × H210 mm area, with degradation in the fringe area of less than 10% compared with the center value.

From here the magnetic structure designed for controlling the bias field of the MI sensor is explained. As shown in the previous explanation of this paper, the MI sensor needs a particular bias field for operation. In the case of our sensor, which is made of Co_85_Nb_12_Zr_3_ amorphous thin film, the bias field is roughly 17 Oe (1.35 kA/m). In order to utilize a high-sensitivity magnetic sensor having a tolerance against the strong field directed in the substrate’s surface normal direction of the thin-film MI sensor, the bias field is indispensable. In this section, the concept of the formation of the bias field structure in this system, which is suitable for a sensor array configuration having multiple sensors in a line at intervals, is introduced.

The method of bias field adjustment was introduced in the previous section in this paper, which was carried out using the sensor position controlling mechanism along the tangential direction of the surface of the homogenization plate. The sensor driver circuit in our proposal made it possible to adjust the sensor position mechanically in the strong magnetic field, which is more than 20 times larger compared with the bias field. The magnetic field generating apparatus was designed for this purpose. It has a distributed vertical field just above the middle law of the magnet array. The distribution has a partial vector along the Y-direction which changes as a function of the Y-position. If this distribution has a particular range of magnetic field, the bias field control is realized.

For the purpose of verifying the bias field generation in this system, magnetic field simulation was carried out due to the difficulty of the actual measurement of it inside the vertical strong field.

Figure 11 shows the compared results of the measurement and simulation. This is the profile of magnetic flux density B_z_ as a function of the feeding directional Y-position. The measurement is the same result as is shown in Figure 10. It shows the cases with the vertical positions *z* = 38 mm and *z* = 93 mm. The simulated results are shown by dashed lines and the measurements are shown by solid lines. The simulation was carried out using Maxwell 3D (ANSYS, Inc., Canonsburg, PA, USA). The results are in good agreement. Therefore, the simulation is a reliable estimate of the field distribution in the vicinity of the homogenization plate. The sensor element is designed to be set at *z* = 5 mm in this system due to a mechanical restriction, which is a 5 mm altitude from the surface of the lower homogenization plate. A difficulty in the precise measurement of magnetic flux density comes from the difficulty in accessing the Hall probe to the surface of the homogenization plate in a strong magnetic field with a strict position arrangement against the coordinate axis.

Figure 12 shows the simulated results of magnetic flux density at the sensor height of *z* = 5 mm. The horizontal axis represents the Y-position, which is the feeding direction. The original point is the central position of the homogenization plate. In this system, the sensor is able to move mechanically, in parallel with the Y-axis, by keeping the sensing direction in the Y-direction using a sliding plate onto which is mounted a sensor element. Figure 12 shows that the magnetic flux in the Y-direction linearly changes between *y* = −40 mm and *y* = +40 mm, and the variation range corresponds to the bias field from B_y_ = −20 G (−2 mT) to +20 G (+2 mT). It also shows that this controlling field is kept within the transversal position range, with *x* = ±300 mm. Based on Figure 5, the parameter *x* = 0 corresponds to the middle position between the plate magnets, the *x* = 150 mm corresponds to the position 30 mm from the center of a plate magnet, and the *x* = 300 mm corresponds to the position at the center of a plate magnet. This variation range for B_y_ is suitable for applying the sensor bias field to our sensor element, having the bias point at about 17 Oe (17 G (1.7 mT) in air circumstances). In this case, it is possible to set the bias field at a suitable value when the position of the sensor element is set at about *y* = 30 mm. The actual fabricated magnetic structure has the possibility to include material non-uniformity and also to include dimensional and attachment mechanical error. Even in this case, our proposed system can adjust the bias point by controlling the sensor position by monitoring the sensor output signal that indicates sensor impedance.

Figure 13 shows the variations in flux density in the transverse X-direction with B_x_, as a function of the feeding Y-position in the range of the bias controlling line. The B_x_ field is in the transverse direction to the sensing direction. If it is the in-plane transverse direction of the thin-film element, then a particular weakness exists against the magnetic field in this direction. Our sensor element is capable of tolerating transverse fields of more than 50 Oe (50 G (5mT) in air conditions). This simulated result shows that the sensor element is able to set and work within the range *x* = ±300 mm.

Based on the results in this section, the proposed magnetic field generation apparatus is capable of applying a suitable bias field to the area of the sensor array installation. The bias point can be set by controlling the Y-position of each element aiming at a particular, suitable magnetic bias field by monitoring the sensor output. The range of the controlling field is designed as a suitably larger value of the bias field in order to include the uncertainty of the material’s non-uniformity and also dimensional and attachment errors.

### 3.3. Verification of Sensor Performance

In this section, an experimental verification of the measurement system in this paper is reported. A single thin-film magneto-impedance sensor was installed in the magnetic field generating structure reported in this paper, and it confirmed the validity and operational performance of the system.

The thin-film magneto-impedance sensor and driver circuit which were used in this study were the same ones which were reported previously by us [22]. A single sensor element was installed for the purpose of confirming the functional performance, although the magnetic field generation system was designed for installing multiple-array sensors on a particular bias field transversal line. We used this system to attempt to detect a millimeter-sized steel chip mounted on the surface of the complex-shaped cast aluminum work piece. Based on recent progress in medical technology, it is possible to now include conductive metal materials in the human body which are used for replacing the functions of the human body. Titanium has been used for this purpose due to its biocompatibility. Examples of such applications are as follows: artificial joints and bone, surgical bone plates and bolts and stents and implants. Although titanium possesses very slight paramagnetic properties, it is regarded as a conductive non-magnetic material. A conventional metal detector is well known to be strongly affected by conductive material. An alternating magnetic field is used for this conventional method, and a small magnetization in the vicinity of the non-magnetic conductive material causes difficulties due to the eddy currents in the conductive material. The eddy current generates strong noise in the detection signal, although the system proposed in this study can detect a magnetized piece in the vicinity of such conductive material. In this study, in order to simulate a cluster of magnetic nanoparticles in the vicinity of a complexly shaped piece artificial bone, we tried to detect a small magnetic chip placed on the surface of an aluminum casting. This is an initial study of our proposed detection system. Further study using an actual bio-medical sample will be carried out as the next steps.

Figure 14 is a photo of the measured small steel chip. It has a 2 mm length and 1 mm width and an ellipsoidal shape resembling the shape and size of a sesame seed. It was mounted on the end-flange surface of a cast aluminum work piece, as is shown in Figure 15. The work piece was fed through the measurement area using the feeding table of this system. The distance between the sensor element and the steel chip was 27 mm in the vertical direction, which was a just above nearest position between them during feeding.

Figure 16 shows the photo of the measurement. The flange surface mounting the steel chip was set as it was placed at the bottom of the work piece, and it was mounted on the feeding table. The work piece was positioned such that the chip was ran through just above the sensor element. The dimensions of the aluminum work piece were a height of 200 mm and a width of 380 mm.

Figure 17 is a typical measurement result without a magnetic chip. The aluminum casting without the chip was mounted on the feeding table. The table started moving at the time *t* = 0, where a step-up signal was detected, and it stopped at time *t* = 105 s, where a step-down signal was detected. The feeding speed in this study was 10 mm/s. The sensor can detect 1 mG in the feeding direction as 0.1 V output. The measured result in this figure shows an increasing tendency as a function of time, and then as a function of the table position. The reason for this increasing tendency was the existence of a steel plate at the bottom of the feeding table that was used to fix it mechanically to the feeding actuator. The variation range was almost 2.3 V of sensor output, corresponding to a 23 mG variation in static magnetic field in the sensor’s sensing direction.

Figure 18 shows the result when the small steel chip was set on the work piece. It clearly detected the existence of the millimeter-sized small chip at a distance of 27 mm from the sensor element using the proposed system in the study. The system can detect it even in the vicinity of non-magnetic conductive bulk metal.

Figure 19 is a display hard-copy of the measured digital oscilloscope. Figure 18 is a direct plot of its digitized and memorized sampling data. It is understood that the detection was clearly carried out, even though there was no application of any post-filtering procedures for eliminating noise from the measured signal. A consideration of the cause and level of the noise in this measurement and the detection limit are explained in the following discussion section.

## 4. Discussion

In this section, the target specifications of the system developed for medical applications is discussed.

From the view point of sensor sensitivity, a prior study utilized a Hall probe which had a sensitivity of 1 μT for detecting magnetic nanoparticle concentrations [29]. It achieved the detection of 140 μg of iron which was contained in 5μL of a fluid of “superparamagnetic iron oxide nanoparticles” at a distance of 10 mm. It was detected by applying a 50 mT static magnetic field to magnetize the nanoparticles. It utilized a special magnetic structure which could install the Hall probe at the magnetic null point. Based on this prior study, the proposed system in this paper has an advantage in sensitivity. The applied static field in this paper is 37 mT, about 3/4 of the prior study’s. In the measurement system which detects magnetic particles using a magnetic sensor simultaneously with magnetization in a static magnetic field, there is a relationship between the magnetization strength and the detection sensitivity of the particle. Considering this effect, an improvement in the sensitivity of the sensor is needed for our sensor system to put into practice the position and size estimation. In order to improve the sensor’s sensitivity, noise reduction in the sensor’s driver circuit is needed.

Regarding the size of the sensor element, the smaller element size has higher precision in position and size estimation, especially when the detected particle is becoming nearer to the sensor element. The reason is the spatial distribution of the detected magnetic field coming from the magnetized particle. The 1 mm sensor element in this paper has a disadvantage from this view point. A newly developed, small size magneto-impedance sensor would be applicable [30]. Additionally, the development of an estimation method for a precise magnetic field using a large sensor element would be considerable. The magneto-impedance sensor has the ability to detect magnetic particles tens of microns in size [19], which are feeding along in the vicinity of the element. There is a particular possibility for developing a method which can estimate a magnetic field as if it were detected by a point sensor using a magneto-impedance strip–shaped element.

From the view point of the realization of the sensor array using multiple sensors, the development of a small unit of the sensor’s driver circuit having a low noise and high-sensitivity is indispensable. A fundamental concept of the driver circuit was already proposed, as shown in this paper, and thus an effort towards miniaturization is the point of future development.

The sensor’s driver circuit which was used in this experiment was the same one as in the previous report [20,21,22]. The uncertainty of the measurement was discussed in the previous paper, in which the circuit noise waveform was shown in Figure 16 in Section 3. Discussion of Ref. [20]. A 0.2 V periodically repeated dipping noise was present with the time interval of 0.67 s. The same figure is reposted in Figure 20 in this paper. This noise is assumed to come from the 400 MHz oscillator, and would be erased by a more sophisticated circuit design. The measurement noise, which is shown in the actual display of the oscilloscope, as shown in Figure 19, was the same level as that in Figure 20. Therefore, by refining our driver circuit, the detection level of the magnetic small piece is expected to be well improved. It is the subject of a future study.

The detection limit of this measurement system is discussed as the final aspect of this paper. It is easily understood that the limit is determined by the sensor sensitivity and also the strength of the magnetic dipole, with the latter having the same meaning as the magnetic moment of the measured sample. The magnetic moment is derived from a multiplication of the magnetization and the sample volume. It is also affected by the distance between the magnetic moment and the sensor. With increasing distance, the strength of the magnetic field coming from the magnetic moment at the sensor position decreases. In order consider the system sensitivity, we have to consider three parameters: sensor sensitivity, the magnitude of the magnetic moment and the detection distance. The fundamental equation for this consideration is a simple one, which was already discussed in our previous work [19,23]. Figure 21 is one of the reposts of this work [19]. It shows a variation in magnetic flux density at the sensor element position as a function of feeding height. The magnetization was assumed as 1 T, and the diameter of the sphere particle was a parameter. Based on this consideration, a *ϕ* 20 μm magnetic sphere can be detected by a sensor having a 10^−8^ T sensitivity at a distance of 3 mm. In order to expand the consideration for the case of a cluster of nanoparticles, we have to consider the dispersed magnetic particles in a particularly shaped volume. Dispersion of the particle concentration inside the volume also needs to be considered, and these are expected as subjects of future study.

## 5. Summary

A uniform and strong magnetic field generator was developed which is applicable to a human body scan. A sensor system in combination with a uniform strong magnetic field and also an array of thin-film magneto-impedance sensors within the strong magnetic field was designed, fabricated, and experimentally evaluated. The magnetic field in the measurement area was uniform in strength and was directionally aligned with a structure having adequate volume to insert a human body. The thin-film sensor was set in the vicinity of the magnet with its sensing direction perpendicular to the strong magnetic field. This system has the ability to detect a stray field of magnetic particles and also to estimate their 3D position and concentration. As a result, a prototype of a magnetic field generator with an area of W500 mm × L400 mm × H210 mm and with a uniform magnetic field of 370 Gauss was developed, and a thin-film magneto-impedance sensor was installed in it. Experimental verification showed that the 2 mm × 1 mm ellipsoidal small magnetic chip, which generated about 1 mG (10^−7^ T) at the sensor, was detected at a distance 27 mm from the sensor.

## Figures and Tables

**Figure 1 sensors-22-03120-f001:**
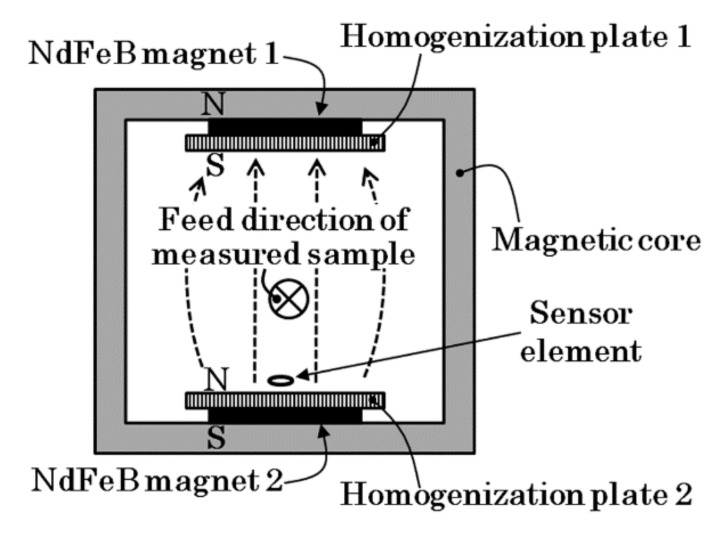
Schematic image of proposed inspection system. (The dotted arrow explains a generated magnetic field).

**Figure 2 sensors-22-03120-f002:**
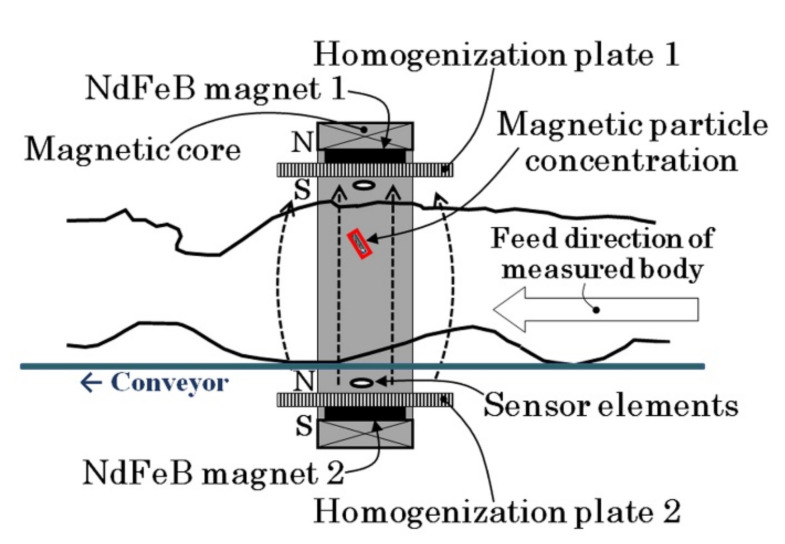
Image of the measurement of a human body using the proposed inspection system.

**Figure 3 sensors-22-03120-f003:**
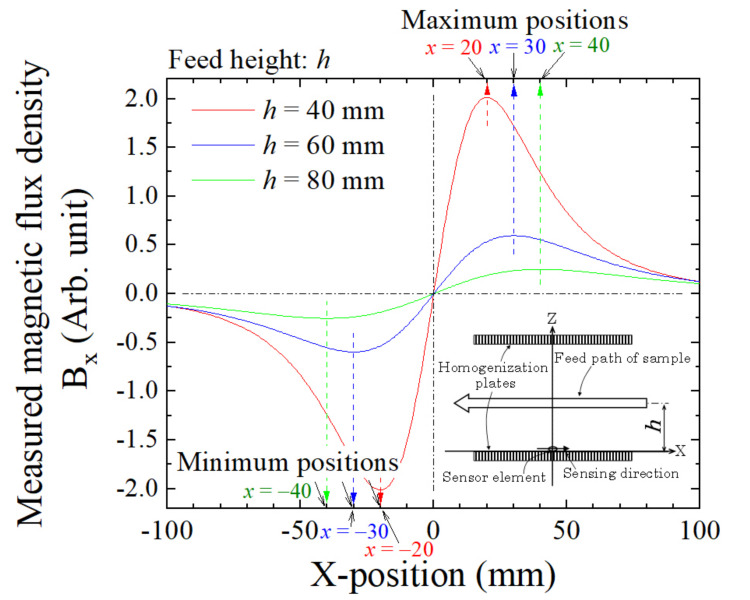
Relationship between the feeding height of a magnetic dipole and the position of an extreme point of measured magnetic flux density [23].

**Figure 4 sensors-22-03120-f004:**
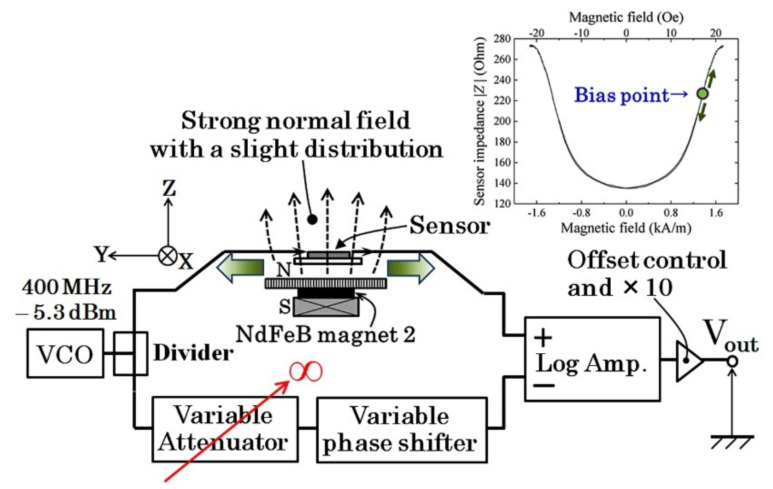
Schematic explanation of the bias field adjustment using a field distribution.

**Figure 5 sensors-22-03120-f005:**
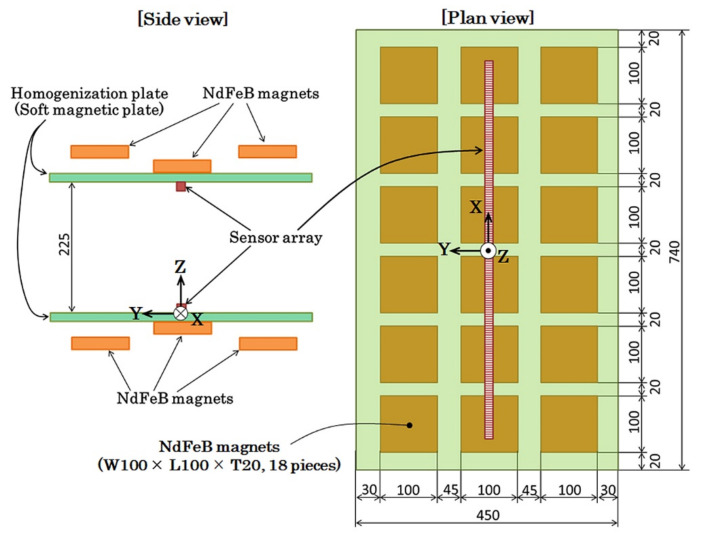
Design layout of the proposed field generation apparatus. (Unit: mm).

**Figure 6 sensors-22-03120-f006:**
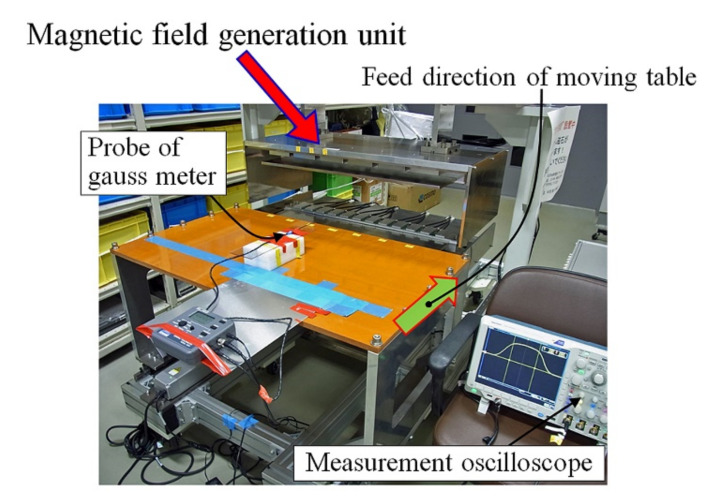
Perspective view of the fabricated measurement system.

**Figure 7 sensors-22-03120-f007:**
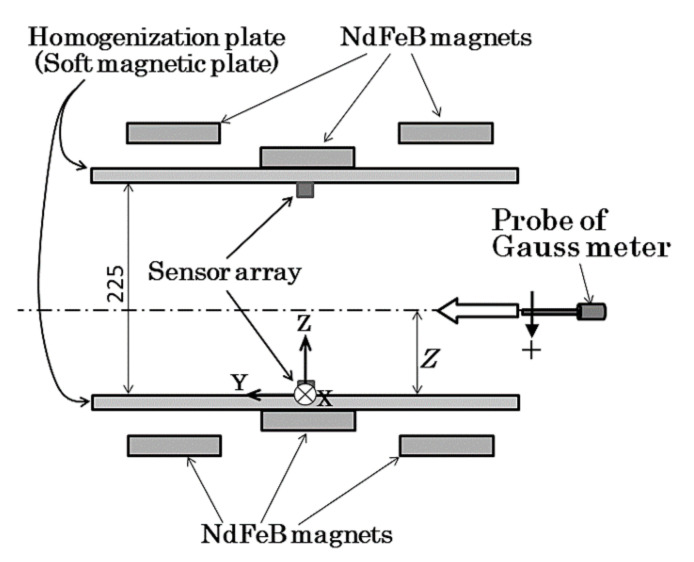
Schematic of the measurement procedure of the magnetic field profile. (Unit: mm).

**Figure 8 sensors-22-03120-f008:**
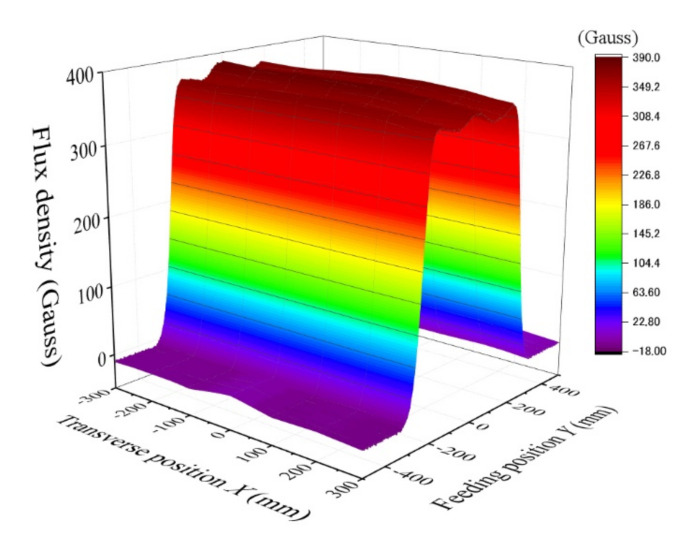
Measured magnetic field B_z_ at the vertical position *z* = 38 mm.

**Figure 9 sensors-22-03120-f009:**
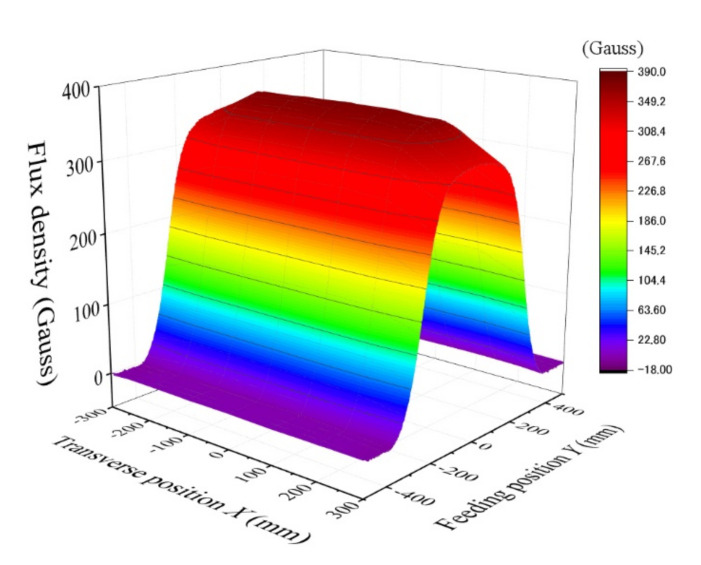
Measured magnetic field B_z_ at the vertical position *z* = 93 mm.

**Figure 10 sensors-22-03120-f010:**
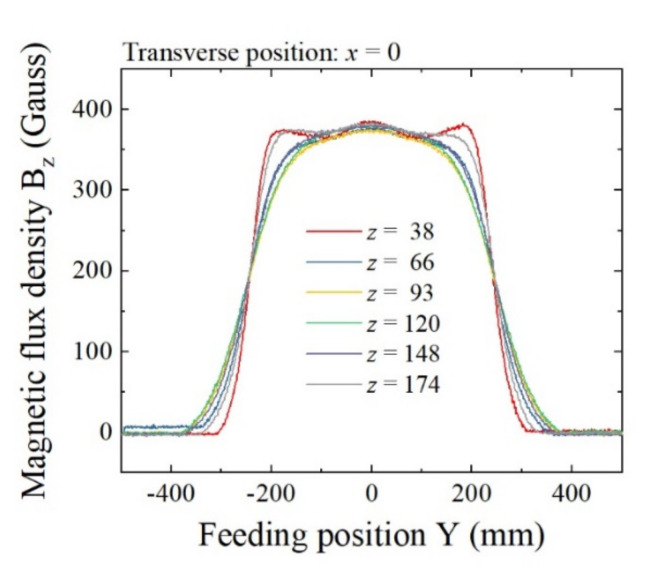
Overlapping expression of the field variation B_z_ on feeding position Y as a parameter of vertical height Z. (Unit of *z*: mm).

**Figure 11 sensors-22-03120-f011:**
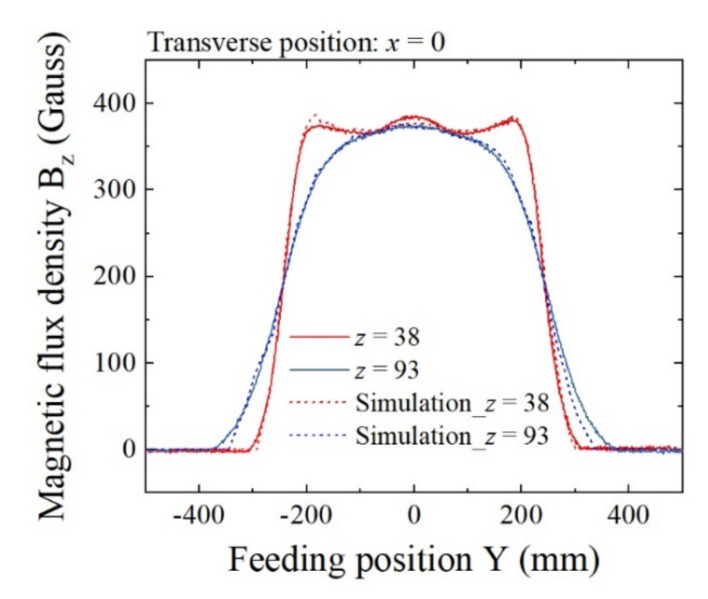
Comparison of measurement and simulation of B_z_ at different heights Z. (Unit of *z*: mm).

**Figure 12 sensors-22-03120-f012:**
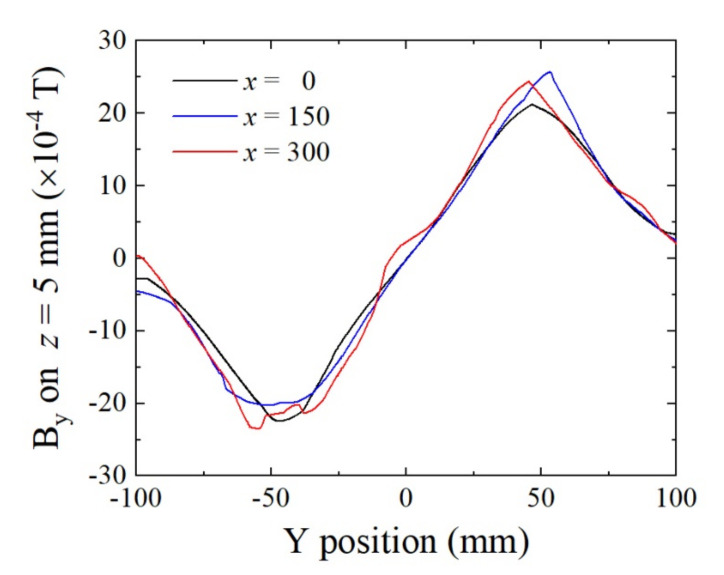
Simulated variations in magnetic flux density B_y_ on Y-position at the sensor height *z* = 5 mm as a parameter of transverse position X. (Unit of *x*: mm).

**Figure 13 sensors-22-03120-f013:**
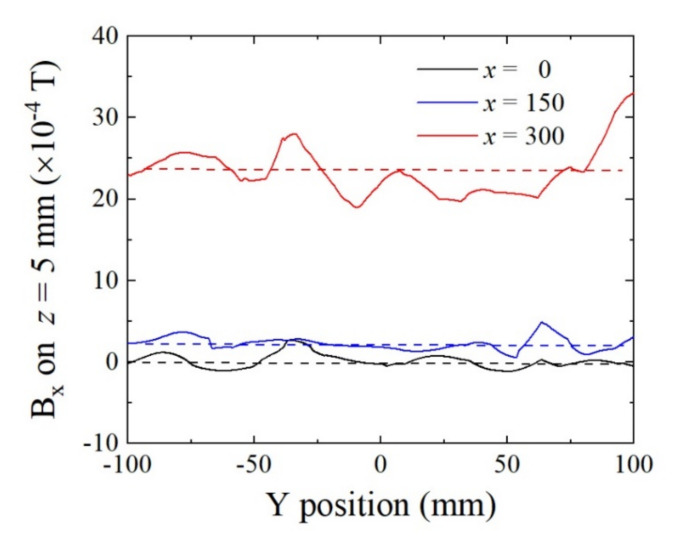
Simulated variations in transverse magnetic flux density B_x_ on Y-position at the sensor height *z* = 5 mm as a parameter of transverse position X. (Unit of *x*: mm).

**Figure 14 sensors-22-03120-f014:**
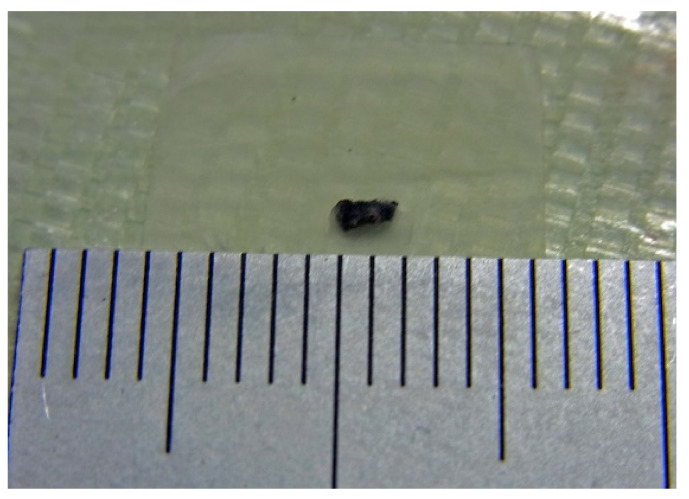
Photograph of the measured small steel chip.

**Figure 15 sensors-22-03120-f015:**
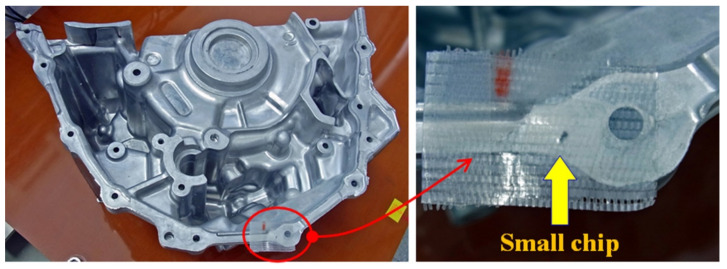
Photograph of the small steel chip mounted on the end-flange of an aluminum casting with a complex shape.

**Figure 16 sensors-22-03120-f016:**
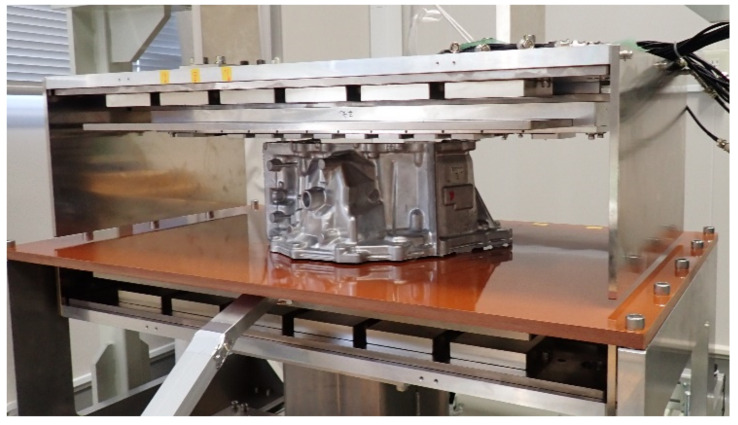
Photograph of the measurement using the fabricated unit.

**Figure 17 sensors-22-03120-f017:**
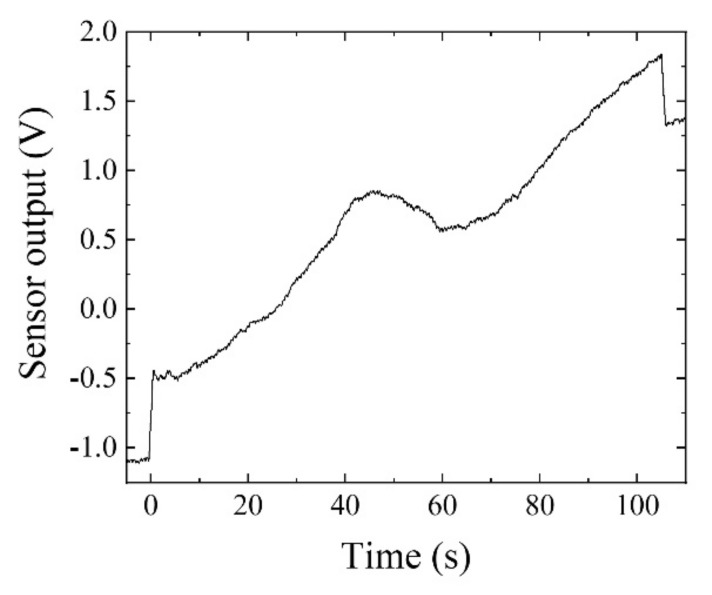
Typical measurement result without magnetic chip. (Feeding speed: 10 mm/s).

**Figure 18 sensors-22-03120-f018:**
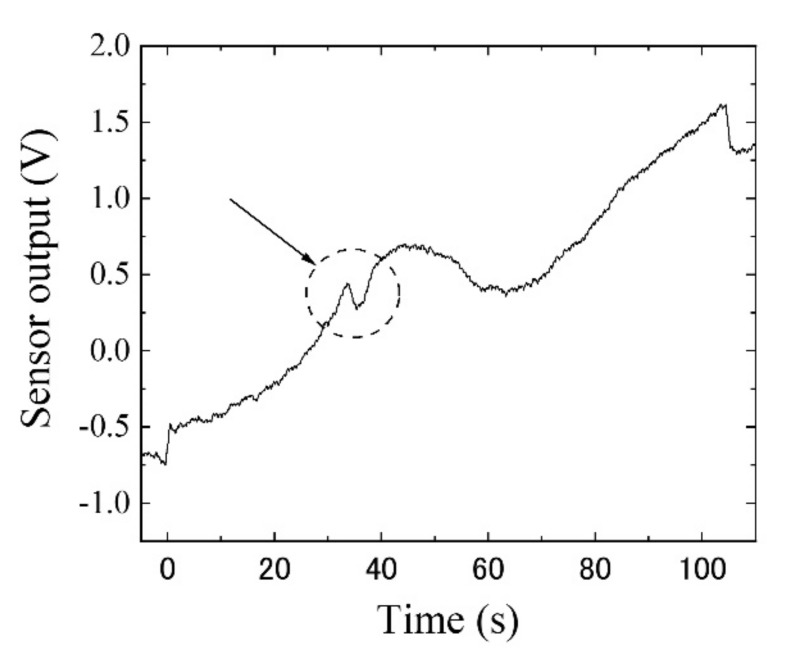
Measurement result when the small steel chip was set on the work piece, as shown in Figure 15. The dotted circle indicates the signal of magnetic chip. (Feeding speed: 10 mm/s).

**Figure 19 sensors-22-03120-f019:**
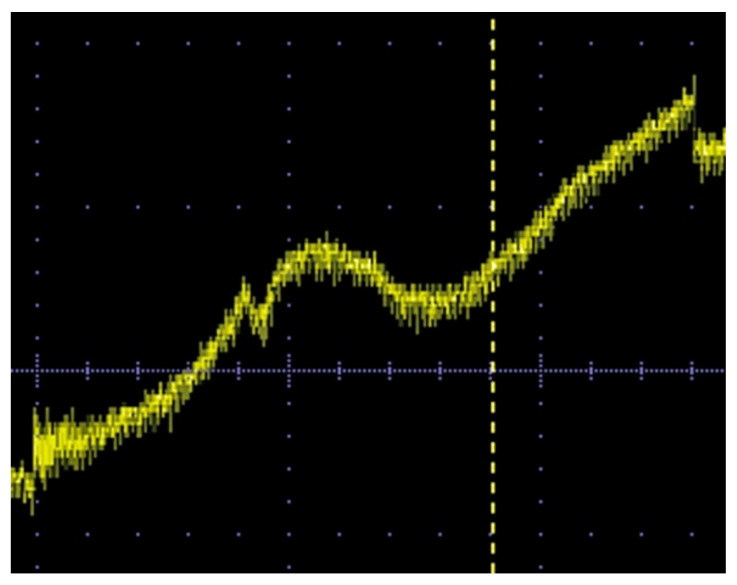
Display hard-copy of the measured digital oscilloscope, during the measurement of Figure 18.

**Figure 20 sensors-22-03120-f020:**
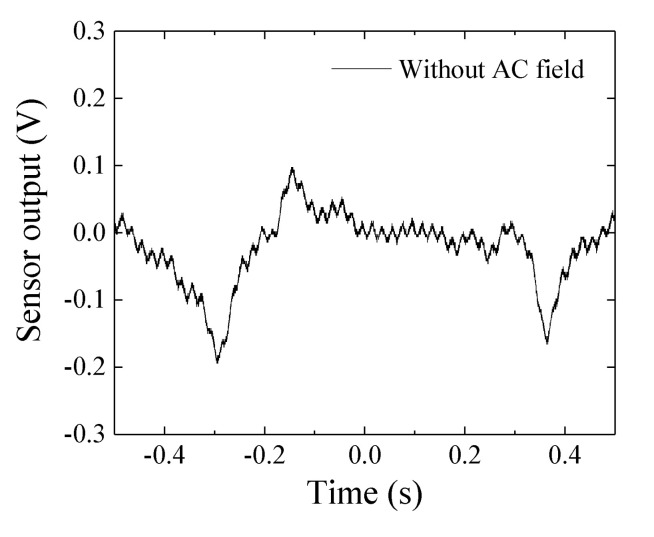
The 0.2 V periodical dipping noise of the sensor’s driver circuit [20].

**Figure 21 sensors-22-03120-f021:**
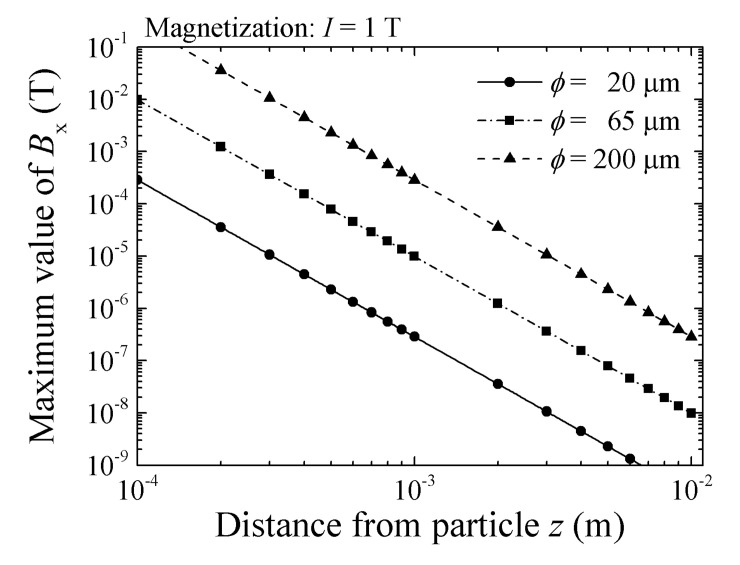
Variations in magnetic flux density at sensor element position as a function of feeding height *z* [19]. (The magnetization of particle: 1 T. The diameter of the sphere particle *ϕ*: parameter. The configuration of measurement system is the same as this paper).

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
