# Peer review of "A Uniform Magnetic Field Generator Combined with a Thin-Film Magneto-Impedance Sensor Capable of Human Body Scans"

_sensors, 2022, doi:10.3390/s22093120_

Round 1
Reviewer 1 Report
“Uniform magnetic field generator combined with thin film magneto-impedance sensor capable of human body scan”, Tomoo Nakai.
The paper presents a GMI-based system for magnetic nanoparticle scanning inside a human body.
The paper is complete and well illustrated.
The referee is not a native English speaker but it seems that some sentences should be re-written to make the argumentation clearer. The paper should be corrected by an English native speaker.
For example:
Lines 48-50: “A recent progress in medical technology utilizes a magnetic nano-particles to be able to introduce in a human body, then a detection of a locally concentrated particles such as in a cancer or in a lymph node is get-50 ting to be important for the medical application. “
Lines 79-81: “This system has an advantageous (advantage) for detecting low remanence magnetic particles, due to (the fact that ?) the sensor measures the magnetic particles simultaneously with magnetization against them.”.
Lines 104-107: “This longitudinal dimension would be possible to decrease (could be decreased), if there is sensor array in both sides of measurement area, which are on the bottom and the upper sides of the magnetic homogenization plate.”
Lines 122-125: “An electrically adjustable attenuator is set in our circuit and makes it possible to block the operation of differential amplification by making the attenuation maximum, then the output indicates the sensor impedance value, and it is possible to adjust the sensor position for fitting the bias point.”
Lines 130-133:” The magnetic field distribution would better be designed to appear along the sensor sensing direction, which is matched to feeding direction of the system, then the sensor bias point is possible to adjust by controlling the sensor position in the feeding direction.” Sentence not clear.
Line 369: “Figure 15. Photo of the measured (what ?) using the fabricated unit. 369”. Measurement set-up, measured sample…
Lines 430-432: “A combination of both the uniform in strength and directionally aligned static magnetic field in a certain measurement area and also the array of single dimensional thin-film magneto-impedance sensor was fabricated as a sensor system and evaluated.”. Not clear.
Various comments:
Page 1, abstract, line 20: express the unit W500×L400×H210 mm.
Page 3, lines 90-95: The author should refer to previous work to explain the significance of H, H’ or detail it here in the paragraph and illustrate it in the figures.
Page 11-13, result section : Why do the author use an aluminum part to evaluate the set-up and not something more representative of a human body which is the target goal ?
References: the author, if possible, should refer to journal papers rather than conferences papers. Journal papers are easier to access for readers.
For example:
Ref [15] : Horia Chiriac, Dumitru-Daniel Herea, Sorin Corodeanu, Microwire array for giant magneto-impedance detection of magnetic particles for biosensor prototype, Journal of Magnetism and Magnetic Materials,Volume 311, Issue 1, 2007, Pages 425-428, ISSN 0304-8853.
Ref [17] : K. Fodil, M. Denoual, C. Dolabdjian, A. Treizebre, and V. Senez, In-flow detection of ultra-small magnetic particles by an integrated giant magnetic impedance sensor, Appl. Phys. Lett. 108, 173701, 2016, doi.org/10.1063/1.4948286
Author Response
Please see attachement

Reviewer 2 Report
The manuscript discusses the thin-film magneto impedance sensor that was installed in a uniform and strong magnetic field. The author verified the detection of an ellipsoidal small magnetic chipping (area = 2 mm2) at a distance less than 30 mm from the sensor element. Overall the results described in the manuscript are interesting.
I only have a few remarks:
1- The following rows have missing units of measurement (mm):
- Abstract, row 20: (W500 mm×L400 mm×H210 mm)
- Concept of measurement system, row 104: (W700 mm×L400 mm×H225 mm)
- Experimental results, row 168: (W900 mm×L120 mm×H12 mm)
- 3.2. Magnetic field distribution, row 204: Z=38 mm
- row 215: y=±225 mm
- row 240: z=38 mm to z=174 mm
- row 246: (W500 mm×L400 mm×H210 mm)
- row 277: z=38 mm and z=93 mm
- row 293: z=5 mm
- row 297: y=+40 mm
- row 301: x=150 mm
- row 302: x=300 mm
- Summary, row 434: (W500 mm×L400 mm×H210 mm)
2- In the following figures, units (mm) are missing:
- Figure 4. In the caption, please add Numbers in mm.
- Figure 6: For 225, please use mm as the unit.
- Figures 9 and 10: In the caption, please put: z (mm)
- Figures 11 and 12: Please in the caption put: z=5 mm and X=150 mm, X=300 mm
3- References:
- Ref 3. 2001 (Bold)
- Ref 16: 2010 (Bold) and the "doi" is missing.
4- 3.2. Magnetic field distribution, row 256: A more accurate number is 17 Oe (1.3528 KA/m = 1.3 KA/m and not 1.4 KA/m).
5- The uncertainty of measurements of " values" should be added.
Author Response
Please see attachement

Reviewer 3 Report
The manuscript “Uniform magnetic field generator combined with thin film 2 magneto-impedance sensor capable of human body scan” by Tomoo Nakai submitted to Special Issue Sensors and Biosensors Related to Magnetic Nanoparticles of MDPI Sensors describes appropriate study well matching the keywords and objectives of this issue. Indeed, magnetic field sensors for detection of magnetic materials in industrial and medical cases are increasingly growing area of research and applications. The Introduction reveals in a correct way different contributions to magnetic impedance applications. However, it is somehow short for the formulation of the main goal of the present research with objective basis.
It would be nice to add some examples of the magnetic materials with nanoparticles adapted for biomedical applications (ferrofluids, nanodiscs, ferrogels) or at least to mention some required characteristics.
It is also important to discuss if this particular design can work with different kind of magnetic impedance sensitive elements. It is mentioned: “In our proposed system, magnetic sensor is a thin film magneto-impedance sensor driven by 400 MHz high frequency electric circuit [20].” Is it possible to use one of the elements mentioned here – just discuss the point in more details: Rivero, M.A.; Maicas, M.; López, E.; Aroca, C.; Sánchez, M.C.; Sánchez, P. Influence of the sensor shape on permalloy/Cu/permalloymagnetoimpedance. J. Magn. Magn. Mater. 2003, 254–255, 636–638; Vas’kovskii, V.O.; Savin, P.A.; Volchkov, S.O.; Lepalovskii, V.N.; Bukreev, D.A.; Buchkevich, A.A. Nanostructuring effects in soft magnetic films and film elements with magnetic impedance. Tech. Phys. 2013, 58, 105–110; Yang, Z.; Lei, C.; Zhou, Y.; Sun, X.C. Study on the giant magnetoimpedance effect in micro-patterned Co-based amorphous ribbons with single strip structure and tortuous shape. Microsyst. Technol. 2015, 21, 1995–2001).
Figures captions are too short and not informative, they must be extended. For example, Figures 11 and 12 has annotation X= 0, X= 150, X =300 but X itself is not defined here.
The minimum size/volume of the steel chipping should be estimated in order to define the lowest detection limit.
Despite some critical comments, work is very interesting and it requires only a minor revision prior to possible publication.
Round 2
Reviewer 1 Report
Thank you for taking into account the comments.
Sincerely yours.